# Diatoms Dominate and Alter Marine Food-Webs When CO$_2$ Rises

**Ben P. Harvey [1],[*] , Sylvain Agostini [1] , Koetsu Kon [1] , Shigeki Wada [1] and Jason M. Hall-Spencer [1,2]**

[1]   Shimoda Marine Research Center, University of Tsukuba, 5-10-1 Shimoda, Shizuoka 415-0025, Japan; agostini.sylvain@shimoda.tsukuba.ac.jp (S.A.); kon@shimoda.tsukuba.ac.jp (K.K.); swadasbm@shimoda.tsukuba.ac.jp (S.W.); jhall-spencer@plymouth.ac.uk (J.M.H.-S.)
[2]   Marine Biology and Ecology Research Centre, University of Plymouth, PL4 8AA Plymouth, UK
[*]   Correspondence: ben.harvey@shimoda.tsukuba.ac.jp; Tel.: +81-0558-22-6697

**Abstract:** Diatoms are so important in ocean food-webs that any human induced changes in their abundance could have major effects on the ecology of our seas. The large chain-forming diatom *Biddulphia biddulphiana* greatly increases in abundance as $p$CO$_2$ increases along natural seawater CO$_2$ gradients in the north Pacific Ocean. In areas with reference levels of $p$CO$_2$, it was hard to find, but as seawater carbon dioxide levels rose, it replaced seaweeds and became the main habitat-forming species on the seabed. This diatom algal turf supported a marine invertebrate community that was much less diverse and completely differed from the benthic communities found at present-day levels of $p$CO$_2$. Seawater CO$_2$ enrichment stimulated the growth and photosynthetic efficiency of benthic diatoms, but reduced the abundance of calcified grazers such as gastropods and sea urchins. These observations suggest that ocean acidification will shift photic zone community composition so that coastal food-web structure and ecosystem function are homogenised, simplified, and more strongly affected by seasonal algal blooms.

**Keywords:** ocean acidification; benthic diatoms; ecological shift; CO$_2$ fertilisation; turf algae; habitat-forming; algal blooms; marine food-webs

## 1. Introduction

Diatoms are dominant marine primary producers, accounting for ~40% of ocean primary production [1]. Ocean acidification, the alteration of carbonate chemistry due to increased anthropogenic carbon dioxide, has negative impacts on many marine calcifying organisms [2], but the possible effects of this rapid change in surface ocean chemistry is still intriguing world experts. There is a growing consensus on how ocean acidification will affect marine phytoplankton [3,4]. Research into the response of diatoms to ocean acidification has mostly focussed on their growth, productivity, and community composition by using mesocosms or flask culture experiments [5], with predominantly positive effects observed. Projections of the effects of ocean acidification on diatoms suggest that increased availability of CO$_2$ as a substrate for photosynthesis will benefit these algae where sufficient nutrients are available [6], and that these algae may indirectly benefit through reduced grazing pressure [7]. Changes in the ecological balance of life in our world ocean due to ocean acidification is a key focus of current research as it underpins the 'blue economy' [8].

Diatoms are strong competitors for ocean resources when sufficient light and nutrients are present; they often dominate early stages of phytoplankton community succession in coastal ecosystems [9]. Under present-day conditions, large diatoms are often restricted by diffusion gradients as they have lower surface-to-volume ratios than smaller species [10]. In incubation experiments, large centric

diatom species benefit from $CO_2$ enrichment and outcompete smaller diatoms [11,12]. A shift to larger diatoms may increase the trophic transfer of energy to marine animals by shortening food chains [13], and promote production in higher trophic levels. Carbon dioxide-driven shifts towards larger diatoms has been observed in ocean acidification experiments using natural communities [7,11,12,14–17], suggesting that larger diatoms may become favoured in the future [5].

Experiments using natural communities of marine organisms are useful when projecting the impacts of ocean acidification as these help assess cascading effects through trophic levels that can only be assessed when interactions and competition among species are considered [18]. Studies investigating the effects of ocean acidification on natural diatom communities have mostly focussed on plankton (review: [5]), typically performed using closed containers [19]. In recent years, alternative approaches have increasingly been used including in situ and long-term mesocosm experiments (e.g., [16,20]) and the use of natural gradients in $pCO_2$. Volcanic seeps can reveal the long-term ecological responses of communities to acidification, while still retaining natural pH variability and intact ecological interactions. Observations along gradients of carbonate chemistry have revealed both winners and losers in acidified conditions [21–23]. For diatoms, $CO_2$ seep research on planktonic species is lacking, since pelagic communities are advected with currents. However, studies at $CO_2$ seeps have found increases in the photosynthetic standing crop of both epilithic biofilms and microphytobenthic assemblages [24,25].

Here, we assess the effect of ocean acidification on diatoms and food-webs. In the context of ecosystem services on which our society relies, it is important to understand how ocean acidification will affect the marine ecosystem structure and function. In the present study, the diatom *Biddulphia biddulphiana* (J.E. Smith) Boyer became more abundant as $CO_2$ levels increased with increasing proximity to a volcanic seep, and at higher levels of dissolved carbon dioxide it was the main habitat-forming species. We investigated how ocean acidification influenced the abundance, photophysiology, and habitat-provisioning of this large diatom.

## 2. Materials and Methods

### 2.1. Study Site, Carbonate Chemistry, and Nutrients

We assessed the response of the diatom *Biddulphia biddulphiana* along a natural gradient of $pCO_2$ at a volcanic seep off Shikine Island, Japan (34°19′9′′ N, 139°12′18′′ E), which we surveyed seasonally through scuba diving from 2015–2018 [22,26]. Shallow sublittoral rocky substrata were spatially dominated by a mix of canopy-forming macroalgae and zooxanthellate scleractinian corals at our reference site and all around this island, except where $pCO_2$ was higher due to seeps. In these high $pCO_2$ areas, there were algal mats previously described as 'turf algae' [22]. We used five sites along the $pCO_2$ gradient: 'Reference' (mean $pCO_2$: 410 ± 73), which was outside the influence of the $CO_2$ seep; 'RCP 2.6' (mean $pCO_2$: 493 ± 158); 'RCP 4.5' (mean $pCO_2$: 765 ± 159); 'RCP 6.0' (mean $pCO_2$: 971 ± 258); and '>RCP 8.5' (mean $pCO_2$: 1803 ± 1287, Table 1). The sites were termed 'RCP 2.6', 'RCP 4.5', and 'RCP 6.0' in reference to their equivalent Intergovernmental Panel on Climate Change Representative Concentration Pathway (RCP) scenarios [27]. Our >RCP 8.5 site was used to assess the abundance and physiology of diatoms to ocean acidification beyond predicted levels due to human $CO_2$ emissions. The areas used in this study had the same temperature, dissolved oxygen, total alkalinity, and depth [22,28].

**Table 1.** Carbonate chemistry of the reference, RCP 2.6, RCP 4.5, RCP 6.0, and >RCP 8.5 sites at Shikine Island, Japan.

| Station | $pH_T$ | Temp (°C) | Salinity (psu) | $A_T$ (µmol kg⁻¹) | $pCO_2$ (µatm) | DIC (µmol kg⁻¹) | $HCO_3^-$ (µmol kg⁻¹) | $CO_3^{2-}$ (µmol kg⁻¹) | Ωcalcite | Ωaragonite |
|---|---|---|---|---|---|---|---|---|---|---|
| **Reference** | 8.041 | 23.086 | 34.129 | 2281.9 | 409.965 | 2007.341 | 1798.117 | 196.978 | 4.76 | 3.115 |
| | 0.067 | 0.603 | 0.741 | 6.80 | 73.383 | 38.944 | 61.612 | 24.859 | 0.596 | 0.392 |
| **RCP 2.6** | 7.983 | 21.437 | 35.056 | 2282.93 | 493.011 | 2044.255 | 1855.972 | 173.103 | 4.144 | 2.703 |
| | 0.119 | 1.273 | 0.125 | 6.57 | 158.004 | 53 | 81.439 | 32.771 | 0.781 | 0.501 |
| **RCP 4.5** | 7.809 | 22.701 | 34.455 | 2283.32 | 765.545 | 2122.447 | 1973.165 | 126.296 | 3.043 | 1.99 |
| | 0.075 | 0.919 | 0.132 | 18.53 | 158.892 | 27.476 | 38.887 | 15.755 | 0.378 | 0.244 |
| **RCP 6.0** | 7.719 | 22.896 | 34.91 | 2271.84 | 970.706 | 2144.537 | 2008.7 | 106.928 | 2.568 | 1.681 |
| | 0.095 | 0.937 | 0.211 | 3.03 | 257.68 | 33.169 | 43.845 | 17.716 | 0.423 | 0.274 |
| **>RCP 8.5** | 7.529 | 22.072 | 34.723 | 2277.62 | 1803.047 | 2218.975 | 2088.23 | 75.92 | 1.823 | 1.19 |
| | 0.234 | 1.212 | 0.742 | 20.50 | 1287.448 | 82.982 | 82.43 | 33.368 | 0.799 | 0.519 |

$pH_T$, temperature, salinity ($n = 336$), and total alkalinity ($A_T$, $n = 4$) are measured values. Seawater $pCO_2$, dissolved inorganic carbon (DIC), bicarbonate ($HCO_3^-$), carbonate ($CO_3^{2-}$), saturation states for calcite (Ωcalcite), and aragonite (Ωaragonite) are values calculated using the carbonate chemistry system analysis program CO2SYS [29]. Values are presented as mean ± S.D. RCP refers to the representative concentration pathway.

Temperature, salinity, and $pH_T$ were measured using multisensors (WQ-22C, TOA-DKK, Japan) deployed simultaneously at each site for one week in June 2019 ($n = 336$, with measurements taken every 30 min). Each meter was calibrated to pH total scale with a seawater standard and certified reference material (oceanic carbon dioxide quality control; obtained from the Andrew G. Dickson laboratory (Scripps Institution of Oceanography). Total alkalinity samples were collected at each site ($n = 4$), immediately filtered at 0.45 µm using disposable cellulose acetate filters (Dismic, Advantech, Japan), and stored at room temperature in the dark until measurement. Total alkalinity was measured using an auto-titrator (916 Ti-Touch, Metrohm, Switzerland). Carbonate chemistry was calculated using the carbonate chemistry system analysis program CO2SYS [29] by using the measured values of $pH_T$, temperature, salinity, and total alkalinity. Disassociation constants from [30], as adjusted by [31], $KSO_4$ [32], and total borate concentrations from [33] were used.

For nutrients, three water samples were collected from each site (using 125 mL Nalgene polycarbonate bottles; Thermo Scientific, USA), and nutrients were analysed using a continuous segmented flow nutrients analyser (QuAAtro39 AutoAnalyzer, Seal Analytical) following standard protocols [34]. Redfield ratios were calculated as the ratio (in moles) of the carbon (C), silicate (Si), and nitrogen (N) to that of the phosphate (P). Carbon values were based on dissolved inorganic carbon measurements.

### 2.2. Field Survey

Percentage cover of *B. biddulphiana* was assessed using haphazardly distributed photoquadrats (50 × 50 cm, $n = 20–25$ per site) on 17 April 2019 at a 5–7 m depth. In order to estimate the mean percentage cover, each photoquadrat was analysed using ImageJ [35] by overlaying 64 points on a grid, and recording the presence or absence of *B. biddulphiana* at each point.

Between 2017–2019, any observations of fish feeding on or interacting with the diatom mat were noted and when possible, the fish species feeding behaviour was qualitatively recorded by video (TG-5, Olympus, Japan).

### 2.3. Photophysiology and Production

A layer of *B. biddulphiana* (collected from their respective sites by hand in May 2019 from a depth of 5–7 m) was attached to 25 mm GFF filters (Whatmann, Pittsburgh, PA, USA) by briefly using a vacuum pump and filter holder in order to achieve a relatively homogenous and flat surface of diatoms. Three diatom-covered filters were prepared for each site and held in filtered seawater for 30 min for dark acclimation. The maximal quantum yield of electron transport yield (Fv/Fm), maximum light utilisation efficiency (α), and maximum absolute electron transport rate (ETRmax) were measured for

each diatom-covered filter using a Junior-PAM (Walz, Germany). The settings used for the PAM were measuring intensity (6), gain (1), saturation intensity (10), and signal width (0.8).

Diatom net oxygen production and respiration were measured in sealed 45 mL glass containers using four fibre-optic oxygen sensors (Firesting Pyroscience, Aachen, Germany) under light and dark conditions, respectively. Measurements were carried out over a 10 min period with $O_2$ measurements being carried out continuously, with a 30 min period in between light and dark measurements. The assumption was made that respiration rate in the light was at a similar rate to the rate in the dark. Diatoms were placed in filtered seawater (0.45 µm cellulose acetate filter), which was set to the appropriate pH/$CO_2$ via the bubbling of $CO_2$. Measurements were always carried out with one blank container to account for microbial respiration. Diatom biomass was standardised using Chlorophyll *a*, which was extracted with DMF (*N*,*N′*-dimethylformamide). After storage for one day in the dark at −20 °C, extinction coefficients were measured at three wavelengths (663.8, 646.8, and 750 nm), according to [36]. Analysis of the net oxygen production and respiration was performed using the 'RespR' package in R [37].

*2.4. Associated Fauna*

Associated fauna was collected using a scuba diver operated airlift to dislodge and lift samples into a 400 µm mesh net for later analysis. Collections were carried out at four plots (25 cm diameter circular quadrat) in the reference and RCP 6.0 sites during May 2016. The aim of sampling the associated fauna was to assess whether the diatom mat in the elevated $p$CO$_2$ conditions supported a similar faunal community relative to a representative equivalent in the reference $p$CO$_2$ (turf algae). Subsequently, random stratified sampling was used with turf algae being sampled in the reference $p$CO$_2$ site, and diatoms in the elevated $p$CO$_2$ site. Samples were fixed in 70% ethanol prior to sorting and identification. Samples were examined under a dissecting microscope, and organisms were separated from the turf/diatom. Fauna were identified to the highest taxonomic resolution possible, and abundance counted.

*2.5. Statistical Analysis*

All statistical analysis was performed in R (v 3.6.0) [38]. For abundance, the data did not conform to normality (QQ) or homogeneity of variance (Bartlett), and so a non-parametric (Kruskal–Wallis) test was used for assessing differences. For measurements of photophysiology, productivity, and faunal species richness, all data conformed to both normality (QQ) and homogeneity of variance (Bartlett). For PERMANOVA, the data conformed to the test for multivariate homogeneity of group dispersion, assessed used 'betadisper' [39].

## 3. Results and Discussion

The marine centric diatom *Biddulphia biddulphiana* (Figure 1A) is widespread, with records off North and South America, Western Europe, Australasia [40], and now Japan. It is often planktonic [41], but can use extracellular polymeric substances [42] to attach to benthic substrata. We found that in high $CO_2$ conditions it consistently forms mats that are up to several centimetres thick (Figure 1B,C). Monitoring along a natural carbon dioxide gradient off Shikine Island over the past five years (2015–2019) has shown that in areas with high $CO_2$, *B. biddulphiana* mats begin to appear in March–April, reaching their peak in June at depths of 6–8 m below Chart Datum (Figure 2C), and that each year, these mats last until the end of the summer (~late August–September) before being removed by strong wave action during typhoons. A similar bloom of *B. biddulphiana* was reported in tropical coral reefs in Mexico (within the Gulf of California), which formed turf-like mats that covered nearby corals [43].

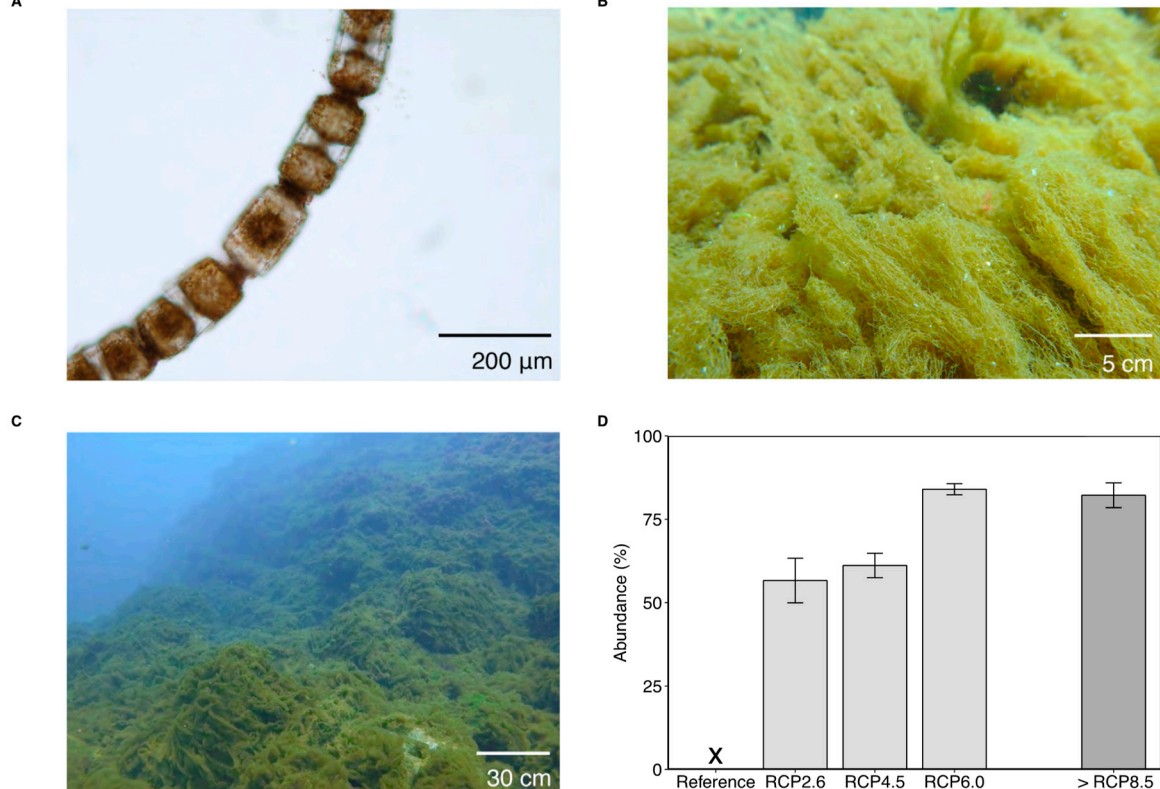

**Figure 1.** (**A**) Chain-forming *B. biddulphiana* diatoms, (**B,C**) these diatoms formed a turf-life mat at our RCP 6.0 sites. (**D**) Percentage cover of *B. biddulphiana*. NOTE: (i) The 'X' in panel (**D**) highlights that the survey was carried out, but zero percentage cover was observed. (ii) The >RCP 8.5 column is shaded and separated to highlight as it does not represent an end-of-the-century projection. Error bars are standard errors.

Benthic diatom mats were never seen by divers during hundreds of surveys at a depth of 0–10 m from 2015–2019 in reference $pCO_2$ conditions. Every year in this five-year monitoring period their abundance greatly increased with increasing $pCO_2$ (ANOVA: $F_{4,114} = 113.5$, $p < 0.001$; Figure 1D). Meta-barcoding of biofilms from a previous study revealed that *B. biddulphiana* were present in reference conditions, but not at high enough abundances for colonies to be visible with the naked eye [44] (May–June 2017, 6–8 m depth). In areas enriched with $pCO_2$, *B. biddulphiana* formed turf-like algal mats (1–10 cm in length, averaging approximately 5 cm in height), which dominated the seabed community (Figure 1). By reaching such a large biomass, and completely covering the seabed, the diatom mat provided a habitat for associated benthos and replaced the canopy forming macroalgae and scleractinian corals that dominated the reference $CO_2$ areas [22]. This suggests that the diatoms benefit from additional dissolved inorganic carbon in the water column and that this allows them to become the main habitat-forming species. Similar boosts in the abundance of diatoms due to elevated $CO_2$ have been shown previously [7,24,25], although this is the first study to show such a large biomass, with previous studies focussing on the microphytobenthos or phytoplankton.

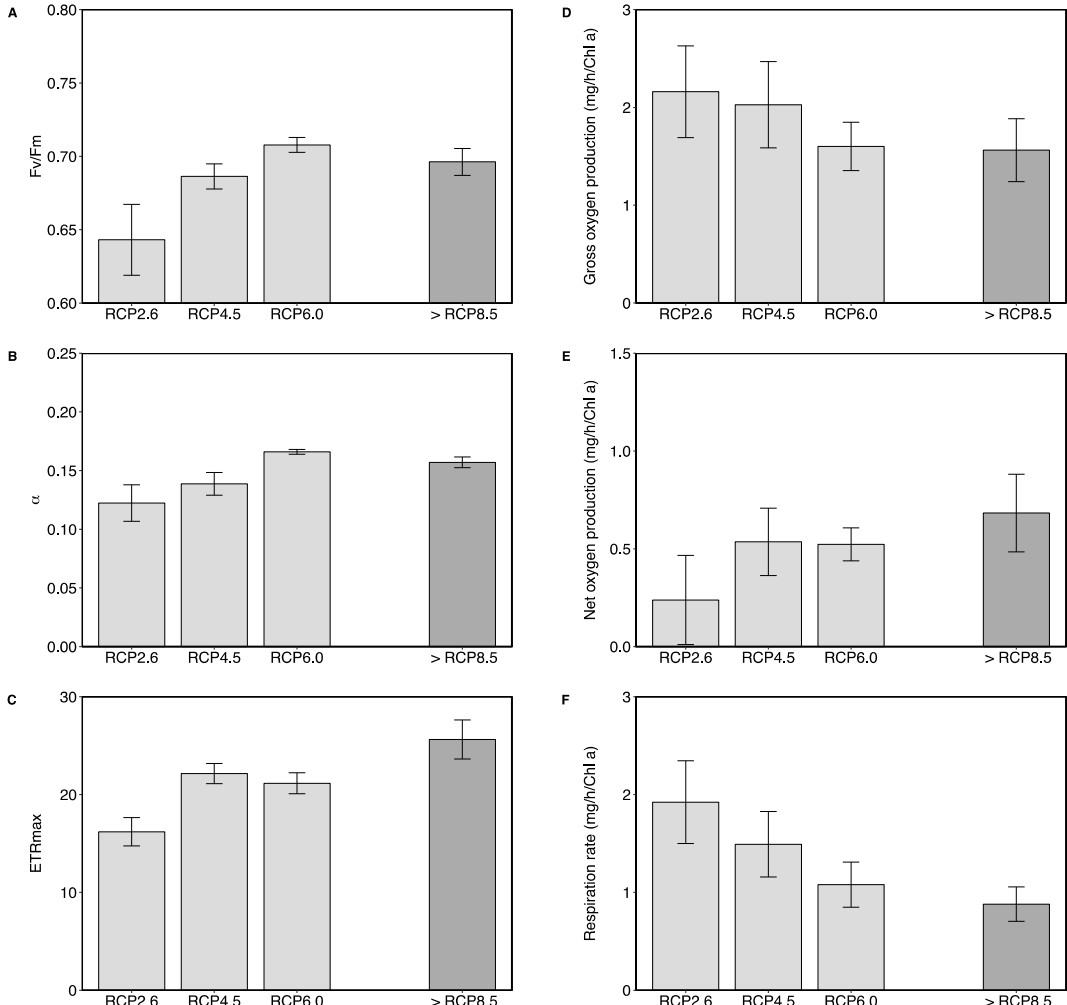

**Figure 2.** Photophysiology (**A–C**), and oxygen production and respiration (**D–F**) of the diatom *B. biddulphiana* under RCP 2.6, RCP 4.5, RCP 6.0, and >RCP 8.5 conditions. (**A**) Maximum quantum yield, Fv/Fm; (**B**) maximum light utilisation efficiency, $\alpha$; and (**C**) maximum absolute electron transport rate, ETRmax. (**D**) Gross oxygen production; (**E**) net oxygen production; and (**F**) respiration rate. NOTE: (i) Panel (**A**) scale starts from 0.6; and (ii) the >RCP 8.5 column is shaded and separated to highlight as it does not represent an end-of-the-century projection. Error bars are standard error.

Diatoms use silicic acid to form their silica frustules. The formation of this silica frustule is a requirement for diatom growth, leading to a strong requirement on silicic acid for growth. In addition, this biogenic silica plays a key role in proton-buffering, benefitting the diatom by facilitating the enzymatic conversion of bicarbonate into $CO_2$ [45]. Based on the Redfield–Brzezinski ratio of 106 (C):15 (Si):16 (N):1 (P) [46], which highlight the typical macronutrient requirements for diatoms, the growth of the diatoms in our site will have been mostly limited by N, and then by P, with Si levels non-limiting across all sites (Table 2 and Figure S1). Since the Redfield–Brzezinski ratio represents a canonical ratio of macronutrient uptake, it is probable that *B. biddulphiana* will have species-specific requirements (depending on its $K_s$, or the half-saturation constant), which could differ from the Redfield–Brzezinski ratio. Studies have not previously established the nutrient uptake kinetic constants for *B. biddulphiana*, highlighting the need for further research. Both nitrate and silicate are considered to be required in broadly equimolar amounts for many diatom species [47], and previous studies have found that the nitrate uptake kinetic constants ($K_s$) for *Biddulphia aurita* [48] are 2.12 ± 0.33 SE μM (strain STX-88 at 25 °C) and 3.19 ± 0.52 SE μM (strain B1 at 25 °C), and that for *Biddulphia sinensis* [49] is 0.74 ± 0.47 SE μM. Constants for *B. biddulphiana* in a similar range would therefore also suggest that nitrate concentrations

at our reference sites (2.58 μM $NO_3^-$ ± 0.22 SD, *n* = 3) and elevated $p$CO$_2$ sites (2.78 μM $NO_3^-$ ± 0.04 SD, *n* = 3) remain limited across all sites. As well as influencing growth and cell maintenance, it is thought that above a threshold of 2 μM, $SiO_4$ diatoms can become dominant within phytoplankton communities [50]. Silicate concentrations were replete at our reference sites (4.62 μM $SiO_4$ ± 0.61 SD, *n* = 3) and elevated $p$CO$_2$ sites (7.79 μM $SiO_4$ ± 0.08 SD, *n* = 3) in the RCP 6.0 conditions), so a lack of silicate likely does not explain the lack of benthic diatom blooms at the reference site and surrounding area. Clearly, diatom community responses will not only be dictated by direct physiological responses to CO$_2$ and macronutrients [5]; they will be indirectly affected by acidification-driven impacts on grazers. Given the high grazing pressure within our reference $p$CO$_2$ site [22,51], predominantly associated with both herbivorous fish and sea urchins, it is possible that *B. biddulphiana* is being excluded. Our results strongly indicate that CO$_2$ enrichment stimulates diatom blooms and when combined with reductions in grazing pressure, allows them to become competitively dominant.

**Table 2.** Nutrient ratios (C:Si:N:P) of the reference $p$CO$_2$, RCP 2.6, RCP 4.5, RCP 6.0, and >RCP 8.5 conditions as well as the established Redfield–Brzezinski ratio for comparison.

| Site | C | Si | N | P |
|---|---|---|---|---|
| Reference | 9050 | 21 | 13 | 1 |
| RCP 2.6 | 10,564 | 39 | 15 | 1 |
| RCP 4.5 | 10,157 | 37 | 14 | 1 |
| RCP 6.0 | 11,049 | 40 | 15 | 1 |
| >RCP 8.5 | 11,469 | 128 | 12 | 1 |
| Redfield–Brzezinski | 106 | 15 | 16 | 1 |

C: carbon, Si: Silicate, N: Nitrogen, P: Phosphorus.

Marine diatoms are dominant marine primary producers [1] and have adapted to modern-day levels of CO$_2$ by operating a carbon concentrating mechanism (CCM), which allows them to elevate the concentration of CO$_2$ at the site of fixation by RubisCO [6]. When levels of seawater CO$_2$ increase, this can stimulate diatom growth through increased photosynthesis and lower energy use though downregulation of their CCMs [52,53]. In our study, enriched CO$_2$ resulted in significant increases in photosynthetic efficiency, in terms of the maximum quantum yield (Fv/Fm: $F_{3,8}$ = 4.17, $p$ < 0.05; Figure 2A), maximum light utilisation efficiency ($\alpha$: $F_{3,8}$ = 4.18, $p$ < 0.05; Figure 2B) as well as maximum absolute electron transport rate (ETRmax: $F_{3,8}$ = 7.34, $p$ < 0.05; Figure 2C). Similar increases in the photosynthetic capacity of diatoms due to elevated CO$_2$ have been previously reported [54–56]. When measuring oxygen production, the gross oxygen production did not show any significant differences between the different sites ($F_{3,8}$ = 0.65, $p$ = 0.62; Figure 2D). Net oxygen production tended to increase with rising CO$_2$ (non-significantly, $F_{3,8}$ = 1.08, $p$ = 0.41; Figure 2E), and this appeared to be driven by a tendency for the respiration rate to decrease under elevated levels of $p$CO$_2$ (non-significantly, $F_{3,8}$ = 2.28, $p$ = 0.16; Figure 2F). Decreases in respiratory metabolism may synergise with the increased rate of C uptake due to increased photosynthesis, promote increased growth rates, and explain the greatly boosted abundance in our site with increasing $p$CO$_2$ levels. Although species-specific responses are likely for diatoms, some generalisations have been suggested, for example, diatoms with lower CCM efficiencies are more likely to show a pronounced response [57], and larger centric diatoms are more likely to profit relative to smaller species [11,12]. This body of work aligns with our novel observations of the effects of ocean acidification on *B. biddulphiana*.

Ocean acidification is expected to simplify communities as stress-intolerant species are lost, and opportunistic species attain competitive dominance [22,58,59]. From a previous study at the site, in reference conditions, a rocky reef habitat had a mixture of both canopy-forming fleshy macroalgae and zooxanthellate scleractinian corals, providing high levels of biodiversity and structural complexity [22]. This shifted to a diatom dominated algal turf community and so we wanted to assess how much associated biodiversity there was in the diatom-based benthic habitat when compared to the reference conditions. The expectation, based on other studies, was that by benefitting the weed-like growth of

just one algal species, this would reduce biodiversity and decrease ecosystem function, secondary productivity, and stability [59–61]. The mobile invertebrate communities that were supported by the diatom mat significantly differed from the communities found in seaweed habitats at reference levels of $p$CO$_2$ (PERMANOVA: $F_{1,7}$ = 6.59, $p$ < 0.05; Figure 3A and Figure S2). They had lower species richness, although this was not statistically significant at our level of sample replication ($F_{1,6}$ = 2.47, $p$ = 0.17; Figure 3B). Several taxa were absent in the elevated $p$CO$_2$ diatom mat (Figure S2), notably the calcified Decapoda and Mysida (Crustacea), Echinea (Echinodermata), and Lucinoida (Bivalvia). Tanaids, which are less calcified, became the most abundant taxon comprising on average ~50% of individuals in elevated $p$CO$_2$ and only ~10% in the reference $p$CO$_2$ turf algae (Figure S2), suggesting competitive release (i.e., a decrease in their predation rates and/or increased availability of suitable habitat) [58,62].

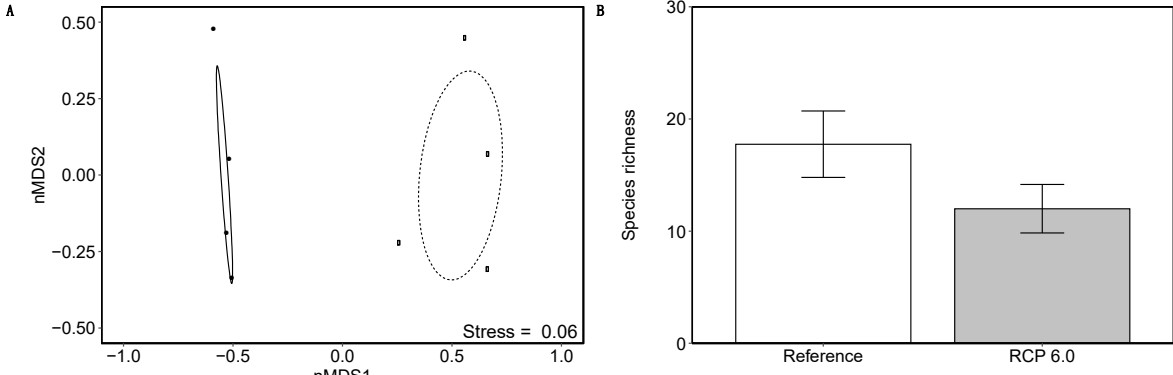

**Figure 3.** (**A**) nMDS mobile invertebrates in the reference $p$CO$_2$ turf-algae (dashed line, open circle) and elevated $p$CO$_2$ RCP 6.0 diatom mat (solid line, closed circle). (**B**) Species richness of the reference $p$CO$_2$ turf-algae (open) and elevated $p$CO$_2$ RCP 6.0 diatom mat (grey). Error bars are the standard error.

Diatoms form the basis of many food-webs, meaning that changes in their growth and/or abundance will have important ramifications for the trophic transfer of energy. In our study sites, the dominant grazers, in both the reference and elevated $p$CO$_2$ conditions, were herbivorous fish (personal observations). Despite herbivorous fishes being present, they did not graze down the boosted biomass of the turf-like diatoms (also see [61]). Only two fish species were seen consuming *B. biddulphiana*, based on ~120 h of in situ SCUBA diving observations, and these only ingested the diatom mat when consuming the invertebrate prey contained within. These were the benthophagous filter-feeding *Cheilodactylus zonatus* (Cuvier, 1830), which uses gill-rakers to capture small invertebrate prey ([63]; and see Figure 4 and Video S1) and the piscivorous and macroinvertivorous *Pseudocaranx dentex* (Bloch and Schneider, 1801), which used ram filtering and suction-feeding on *B. biddulphiana* to consume its prey ([64]; and see Video S2).

Carbon dioxide seeps are open systems that allow recruitment from outside, so while these systems are useful in showing which marine organisms are resilient today, it does not show the potential role that genetic adaptation will have over the coming years [22,60,65]. Regardless, the results of this study can provide important insights into how marine ecosystems could be altered in the near future. Increasing CO$_2$ levels were accompanied by a shift from diverse benthic communities of corals and macroalgae to a diatom turf community. Similar general patterns are seen at CO$_2$ seeps in tropical, sub-tropical, and temperate coastal systems with algal dominance, habitat degradation, and loss of biodiversity in acidified areas [66]. Diatom and turf algal blooms have been observed at other CO$_2$ seeps, for example, at the temperate White Island CO$_2$ seep in New Zealand where turf algae outcompete kelp [61]. Communities composed of simplified, opportunistic species have less ecological stability [67] since they can typically be succeeded by other competitively dominant species [68]. For a simple early successional community to maintain dominance suggests competitive exclusion and/or feedback loops that lock the system into a simplified state [68]. Overall, such shifts are likely to mean

that the coastal food-web structure and ecosystem function will become homogenised, simplified, and more strongly affected by seasonal algal blooms.

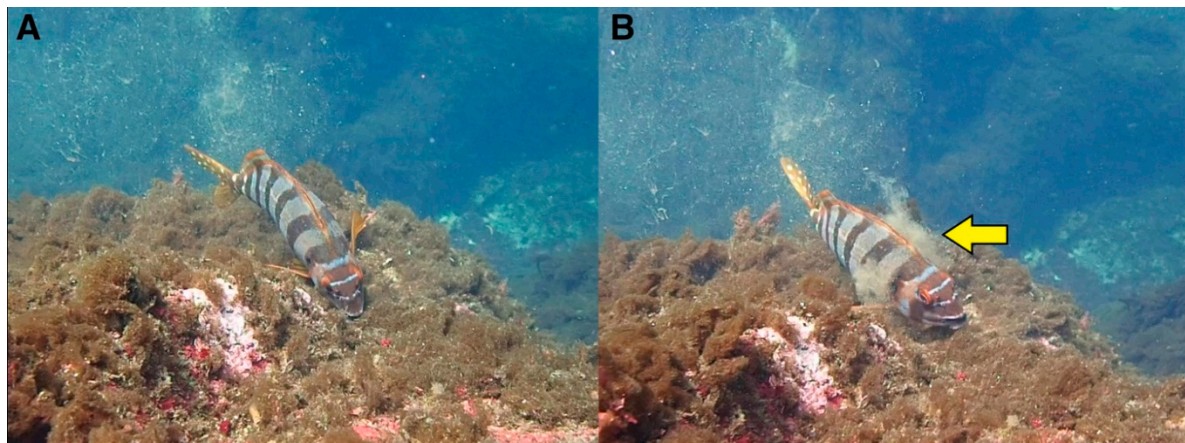

**Figure 4.** (**A**) The benthophagous filter-feeding *Cheilodactylus zonatus* (*Cuvier, 1830*) that uses gill-raking to capture small invertebrate prey from the diatom mat. (**B**) The yellow arrow indicates the remaining diatom being pushed through the gills (see Video S1 for a video of this feeding approach).

In conclusion, the seabed habitat analysed here showed significant changes resulting from $CO_2$ enrichment. Elevated levels of $CO_2$ stimulated the growth and photophysiology of a large chain-forming species of diatom, enabling it to become the dominant benthic habitat-forming species. This thick turf-like algal bloom was capable of supporting an abundant mobile faunal community, although the supported community differed from a typical seaweed community found in the reference $pCO_2$ conditions. Such ecological shifts will have important impacts on food web structure and ecosystem functioning.

**Supplementary Materials:** The following are available online at http://www.mdpi.com/1424-2818/11/12/242/s1. Figure S1: Nutrient concentrations (A–D) of the reference $pCO_2$, RCP 2.6, RCP 4.5, RCP 6.0, and >RCP 8.5 conditions. (A) Nitrite ($NO_2$), (B) Nitrate ($NO_3^-$), (C) Phosphate ($PO_4$), and (D) Silicate ($SiO_4$). NOTE: The >RCP 8.5 column is shaded and separated to highlight as it does not represent an end-of-the-century projection. Error bars are the standard error; Figure S2: Abundance (individuals per $m^2$) of the mobile invertebrate fauna found in the reference $pCO_2$ macroalgal turf (white filled) and elevated $pCO_2$ RCP 6.0 diatom mat (grey filled). Mobile invertebrate fauna are divided at the taxonomic order level, with broader taxonomic phylum groupings indicated. NOTE: Abundance is displayed using log scale. Error bars are the standard error. Video S1: Feeding behaviour of the benthophagous filter-feeding *Cheilodactylus zonatus*, which uses gill-raking to capture small invertebrate prey from the diatom mat; Video S2: Feeding behaviour of the piscivorous and macroinvertivorous *Pseudocaranx dentex*, which employs ram filtering and suction-feeding on *B. biddulphiana* to consume its prey.

**Author Contributions:** Conceptualisation, B.P.H.; Methodology, B.P.H.; Validation, B.P.H. and S.A.; Formal Analysis, B.P.H.; Investigation, B.P.H., S.A., K.K., and S.W.; Resources, B.P.H., S.A., K.K., and S.W.; Data Curation, B.H.; Writing—Original Draft Preparation, B.P.H.; Writing—Review & Editing, B.P.H., S.A., K.K., S.W., and J.M.H.-S.; Visualisation, B.P.H.; Supervision, B.P.H. and J.M.H.-S.; Project Administration, B.P.H.; Funding Acquisition, B.P.H., S.A., K.K., S.W., and J.M.H.-S.

**Funding:** This work was partially supported by Japan Society for the Promotion of Science (JSPS) KAKENHI grant number 17K17622, and the Ministry of Environment, Government of Japan (Suishinhi: 4RF-1701).

**Acknowledgments:** We thank Yasutaka Tsuchiya and the technical staff at the Shimoda Marine Research Centre, University of Tsukuba for their field assistance. This project contributes towards the 'International Education and Research Laboratory Program', University of Tsukuba. We acknowledge funding support from Japan Society for the Promotion of Science (JSPS) KAKENHI (grant number 17K17622), and the Ministry of Environment, Government of Japan (Suishinhi: 4RF-1701).

**Conflicts of Interest:** The authors declare no conflict of interest.

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
