# Peer review of "Diatoms Dominate and Alter Marine Food-Webs When CO2 Rises"

_diversity, doi:10.3390/d11120242_

Round 1
Reviewer 1 Report
This manuscript reports on the response of the benthic diatom Biddulphia biddulphiana along a natural gradient of pCO2 at a volcanic seep off Shikine Island, Japan. The authors show that these natural gradients led to the formation of algal mats of the diatom B. biddulphiana that influenced associated communities of benthic animal. The competitive ability of diatoms under high pCO2 conditions and sufficient nutrient conditions is proposed as the cause of the replacement of seaweeds, dominant at the reference site, by the diatom algal turf, dominant in Representative Concentration Pathway (RCP) scenarios 2.6, 4.5, 6, and 8.5. The study is well designed and the manuscript is well written. The observation has broad implications as the observed RCP scenarios might potentially be representative of many coastal regions in the coming decades. However, I have a major concern with the cause proposed to explain the massive increase of the diatom B. biddulphiana. The authors associate the diatom population dynamics with increasing pCO2 levels but there is also a parallel gradient in Si availability that might potentially contribute to. The authors argue that, according to the Redfield-Brzezinski ratio of 106 (C) : 15 (Si) : 16 (N) : 1 (P), the growth of the diatoms would have not been limited by Si levels but by N and P. They also support their arguments on the view that, in previous mesocosms experiments, diatoms dominate marine phytoplankton community biomass when Si availability exceeds 2-micromolar. However, the Redfield-Brzezinski ratio represents an average (a canonical ratio) with raw data varying around such that diatoms forming thick frustules would require Si concentrations above the Redfield-Brzezinski ratio and vice versa. For instance, the half-saturation constant for Si uptake (Ksi) of southern ocean diatoms, which tend to be heavily silicified, is much higher than the Ksi for subtropical taxa. According to FigS1, there seems to be an excess of both nitrate and silicate along the pCO2 gradient (note that concentrations reflect the balance between supplies and consumption). Therefore, how can we discard the effect of Si supply on diatoms without conducting parallel experiments that allow us to determine the physiological properties and nutritional requirements of the diatom species under scrutiny? At the very least, I would recommend providing data from the literature on the Ksi of the species B. biddulphiana or other diatom species adapted to thrive under Si-saturating conditions to demonstrate that the diatom algal turf was not limited by Si. As stated above diatom limitation could begin at relatively high Si concentrations in the bulk medium - and so high concentrations of Si in the medium cannot be discarded as limiting of diatom growth. The fact that there is an excess 5-micromolar in the reference site where no diatom turf was observed suggests that the diatom R* for Si, the minimum Si requirement below which the species is unable to grow, would be around 5-micromolar.
Reviewer 2 Report
The text is generally well written and easily understandable. However, I have several comments that may help improve the clarity.
Abstract
Statement: “In areas with pre-industrial levels of pCO2 it was hard to find, but as seawater carbon dioxide levels rose it replaced seaweeds and became the main habitat-forming species on the seabed.” Where and how and when did the authors locate areas with pre-industrial levels of pCO2?
Introduction
There is a sweeping overview on the effect of high CO2 on ocean ecosystem functioning. I would expect a little bit more background information on previous findings in comparable sites elsewhere.
The test Biddulphia biddulphiana (J.E. Smith, Boyer 1900) needs to be cited as follows: Biddulphia biddulphiana (J.E. Smith) Boyer, and then if needed, cite Boyer (Boyer 1900)
M&Ms
Some results presented in Results and Discussion seem not to have any methods presented here. For instance, the results of Redfield ratios. I suggest not to clutter text with values one can find in Table 1, to write in legend of Table 1 that RCP refers to Representative Concentration Pathway, and to provide a reference for carbonate chemistry system analysis program CO2SYS. Table- and Figure legends should explain the content. If values in Table 1 are presented as mean ± S.D, then state how many replicate measurements were taken (n=x). Maybe worth to document exactly when in April 2019 measurements were taken in order to allow for future comparison.
Results and Discussion.
Since I found it not always crystal clear what is Results and what Discussion (or Introduction), I strongly recommend that the authors separate the two. This will also clarify what are the results specifically obtained in this study and what has already been published elsewhere. Statement 139-141 seems to me a typical “Introduction” sentence. Lines 162-173 refer to pre-existing information because there are citations. In other words, ambiguity between what has been achieved in this study and what refers to earlier or other work. If earlier work, then I recommend adding this information to the Introduction.
The values in Figure 2B, are they extrapolated from the 10-minute measurements to the entire 24 h cycle, taking into the account the day-night cycle? Figure 2D respiration could instead be extrapolated over 24h. I ask this because it seems that Figure 2F minus Figure 2D is Figure 2B. Or have I misunderstood?
Why is there a gap between columns RCP6.0 and >RCP8.5 in the figures?
I do not follow the argument in lines 173-184, but this maybe because I am not familiar with the methods. The authors mention a Redfield-Brzezinski ratio of 106 (C) : 15 (Si) : 16 (N) : 1 (P) [36] for diatoms and then they show data in Table 2, but it is not clear from the M&Ms how these data were generated or if these data are typical for Bb. I can imagine that there are quite some differences among diatoms depending on the robustness of their silica frustule. Notably, the C levels seem way outside the expectation, even at the reference site. If Si refers to the amount of silica in the cell wall of Bb, then there is quite a stark difference between Reference and >RPC 8.5.
Lines 188-192 state that diatoms have a carbon concentrating mechanism. It is my understanding that such systems have a cutting edge over those that do not have such a CCM .... under low CO2 concentrations because the RubisCO will be less prone to take in O2 instead of CO2. So, I wonder where the cutting edge is under high CO2.
Figures should be numbered in order of appearance in text. For instance, Figures 1C and 1D are mentioned before Figure 1A. There is mention of Figure 3C and 3D in the text but they are nowhere to be found.
I would spend some words on outlook. The work studies the effect of high CO2 (and apparently also elevated Si) on a benthic community near a cold seep and extrapolates the observations to the future high CO2 world. The conclusion should be that high CO2 fosters a fluffy, virtually monotypic diatom community over a diverse benthic community of corals and macroalgae. This can then very tentatively be extrapolated into the future with the caution that the study cannot take into the account adaptation over the coming years. Not to mention that such monotypic stands are prone to adversaries, earlier or later.
I also recommend the authors to compare their results with those of studies at other high CO2-low pH cold seeps elsewhere. For instance: Hall-Spencer et al. (2008) Volcanic carbon dioxide vents show ecosystem effects of ocean acidification. Nature 454:96-99. Did these authors show similar trends in their systems? Was this site also swamped by diatom fluff?
By the way, could it be that the meiofaunal herbivores keeping Bb at bay under normal condition do not like the higher CO2 lower pH environment? To me it sounds quite astounding that a slight difference in photosynthetic performance explains the difference between a virtual no-show of Bb under normal condition versus a monotypic stand under high CO2.
A experiment that seems to be missing is the isolation of diatoms into a series of monoclonal strains and growing these in a range of pCO2 and see how that affects their growth rate. But that could be done in a follow-up study.
Round 2
Reviewer 1 Report
The authors have responded to all my queries satisfactorily.
I am pleased to recommend this manuscript for publication in the journal Diversity.
Kind regards,
Pedro
Reviewer 2 Report
Dear Authors,
I am satisfied with your responses and rebuttals on my comments and content with the edits in the new version of the manuscript. Congratulations.